# Gut microbiota and its influence on the Gut-Brain axis in comparison with chemotherapy patients and cancer-free control data in Breast cancer—A computational perspective

**Tamizhini Loganathan, George Priya Doss C** *

Laboratory of Integrative Genomics, Department of Integrative Biology, School of BioSciences and Technology, Vellore Institute of Technology (VIT), Vellore, Tamil Nadu, India

* georgepriyadoss@vit.ac.in

## Abstract

Breast cancer (BC) continues to be a major cause of cancer-related illness and death among women worldwide. Traditional treatments include surgery, radiation, hormone therapy, and chemotherapy, but these approaches often face challenges due to variability in patient response and adverse effects. This study investigated the relationship between gut microbiome diversity, community composition, and pathway analysis in women undergoing chemotherapy for BC (During Treatment-DT) compared to cancer-free controls (CFC). Using 16S rRNA amplicon sequencing, the study assessed alpha and beta diversity. Results showed differences in microbiome composition between DT and CFC samples, with *Firmicutes* being highly abundant in both groups. Core microbiome and correlation analysis at the phylum and genus levels identified significant microbiota. Specifically, the abundance of genera such as *Pseudomonas* and *Akkermansia* decreased, while *Ruminococcus* and *Allistipes* increased, as determined by statistical and machine learning approaches. Disease associations were examined based on KO abundance, identifying links to conditions such as autism spectrum disorder, Clostridium difficile infection, chronic kidney disease, and multiple sclerosis. Key KEGG pathways enriched in DT and CFC groups included the two-component system, tyrosine metabolism, and the pentose phosphate pathway. Conversely, dysbiosis or the presence of pathogenic bacteria (*Ruminococcus*) associated with the *SOX8* gene could lead to chemoresistance, altered metabolic pathways, and increased toxicity. These findings underscore the potential implications for treatment outcomes and personalized medicine.

## Introduction

Breast cancer (BC) remains one of the most prevalent malignancies among women globally [1]. The incidence of BC exhibits considerable variation across different

**Data availability statement:** The current study data are available in the main manuscript. The link of the files was provided here (https://github.com/Initamizh/Microbiome_treatment_BC.git).

**Funding:** The author(s) received no specific funding for this work.

**Competing interests:** NA.

regions and populations, influenced by genetic predispositions, lifestyle choices, and environmental factors [2,3]. According to the World Health Organization (WHO), BC is the most frequently diagnosed cancer among women worldwide, with over 2.3 million new cases annually [4]. The American Cancer Society projected approximately 297,790 new cases of invasive BC in 2023. Similarly, the European Cancer Information System (ECIS) reported around 355,000 new BC cases in 2020 [1,4]. In Asia, incidence rates vary significantly, with urban areas showing higher rates than rural regions [5]. Multiple factors influence post-chemotherapy BC recurrence rates. Although chemotherapy has markedly improved BC survival rates, it is accompanied by a spectrum of side effects [6,7]. It is crucial to balance the therapeutic benefits and adverse effects with ongoing advancements in treatment protocols to enhance efficacy while minimizing negative outcomes [7]. Side effects are categorized into acute (short-term) and chronic (long-term). Acute side effects encompass nausea, alopecia, fatigue, infections, and weight loss, whereas chronic side effects include cardiotoxicity, peripheral neuropathy, cognitive impairment, fertility issues, and secondary malignancies [8]. This study specifically examines the interplay between gut microbiota and chemotherapy. The gut microbiome plays a pivotal role in modulating the response to chemotherapy and its associated side effects. Research indicates that gut microbiota can influence chemotherapy's efficacy and its side effects severity [9,10]. The gut microbiome can metabolize chemotherapeutic agents, thereby affecting their therapeutic efficacy. Certain bacteria can activate or deactivate specific chemotherapy drugs, altering their clinical outcomes [11].

Additionally, the gut microbiota can modulate the immune system, which is crucial for the body's response to cancer and chemotherapy. A healthy gut microbiome can enhance the immune system's capacity to target cancer cells [12]. Chemotherapy can disrupt the gut microbiome, leading to gastrointestinal side effects such as nausea and mucositis [13]. A disrupted microbiome can exacerbate these side effects. Alterations in the gut microbiota can result in increased systemic inflammation, worsening chemotherapy-induced side effects, and overall patient health [14]. Researchers are investigating various strategies to modulate the gut microbiome to improve chemotherapy efficacy and mitigate side effects [15,16]. The present study analyzes fecal microbiota composition during treatment (DT) and in cancer-free control samples (CFC). Various taxonomic levels are employed to assess microbial abundance. Microbial communities' α and β diversity are evaluated by comparing groups based on richness, dominance, and similarity indices. The differential abundance of microbiota is determined using statistical and machine learning methodologies. Functional enrichment of pathways and disease associations are linked to distinct microbiomes. The gut-brain axis highlights the importance of a holistic approach to cancer treatment, considering the interconnectedness of these systems. Maintaining a healthy gut microbiome through diet, probiotics, and stress management can enhance both physical and mental health outcomes for BC patients. Ongoing research continues to elucidate these interactions, paving the way for more integrated and effective treatment strategies.

## Methods

### Data collection

This study compares and analyses the dysbiosis in the fecal microbiomes of cancer-free control (CFC) and during-treatment (DT) samples. We searched the NCBI BioProject using the terms "breast cancer" and "microbiome," selecting "human" as the organism type and "metagenome" as the study type. Fecal samples obtained from cancer-free control samples and breast cancer patients receiving chemotherapy were sequenced using 16S rRNA technology (Bio-Project ID: PRJEB54599). The samples were single-end, and the sequencing platform is Illumina MiSeq. The samples were analyzed in region V4 with 33 samples. This region is used for taxonomic classification and identification of bacteria due to its sequence variability between different species. It typically ranges from 250 bp. The subjects included two sets of females: one group receiving chemotherapy for stage I-III breast cancer and another group of healthy controls. Further requirements for both categories involved being able to sign and date a consent form, being aged between 18 and 75 years old, and having the ability to read and comprehend English. Both groups were excluded if they had sensory or language impairments, had taken antibiotics in the last 14 days, or had comorbidities like congestive heart failure, liver or kidney disease, COPD, autoimmune disorders, or celiac disease. Excluding individuals with stage IV breast cancer from the BC group was necessary for this research, as their disease had spread and required a different treatment approach [14]. The metadata information of these samples is mentioned in S1 Table. The detailed study of this sequencing, library construction, and processing of the reads were mentioned [14].

### Raw reads processing

The fecal samples of raw FASTQ files were obtained from the European Nucleotide Archive (ENA) [17]. The single-end reads fetched from 16s rRNA sequencing were analyzed using Qiime2 version 2023.5 [18]. The raw data was uploaded to Qiime2. The samples were demultiplexed to check the summary of the data. Every read-sequencing sample has a barcode sequence that identifies the original sample. Demultiplexing could involve quality control procedures to eliminate primer or adapter sequences and low-quality bases and reads. Low-quality reads were removed through trimming and truncation techniques (Q < 30). Trimming was performed at beginning position 0 and truncated at base length 150 bp for the single-end reads. After quality control, the deblur method is used to denoise the readings [19]. Qiime2 creates a feature table that shows the abundance of each unique sequence variant (ASV) across samples after denoising with deblur. Using the pre-trained Naive Bayes approach and the taxonomy categorization, the ASVs were pre-clustered at a 99% sequence similarity criteria using the SILVA database (v. 13_8) [20]. The DNA sequence was matched to a microbial monotype at the phylum or species level using taxonomy analysis. Sequences from the phylum-level mitochondria and chloroplasts and those from the kingdoms Archaea and Eukaryota were eliminated during preprocessing. The resulting QIIME artifacts—the taxonomy table, phylogenetic tree, and feature table—were subjected to statistical analysis. Fig 1 details the entire process, from raw reads to downstream data.

### Bioinformatics data analysis

The downstream analysis was further analyzed through MicrobiomeAnalyst 2.0 [21]. Rstudio programs were used to enter the feature table, metadata, and taxonomy table data in the appropriate format for the MicrobiomeAnalyst tool. The processing and summary data are provided by the data integrity check. Features that are poor in quality, low in abundance and low in variation will be eliminated using the data filtering option. Data normalization is necessary to consider the data's sparsity, undersampling, and unequal sequencing depth. In our study, we have used Total Sum Scaling (TSS) normalization, where each feature count is divided by the total sum of counts per sample to account for differences in sequencing depth. Various downstream analyses are available for marker gene analysis, including clustering analysis, functional

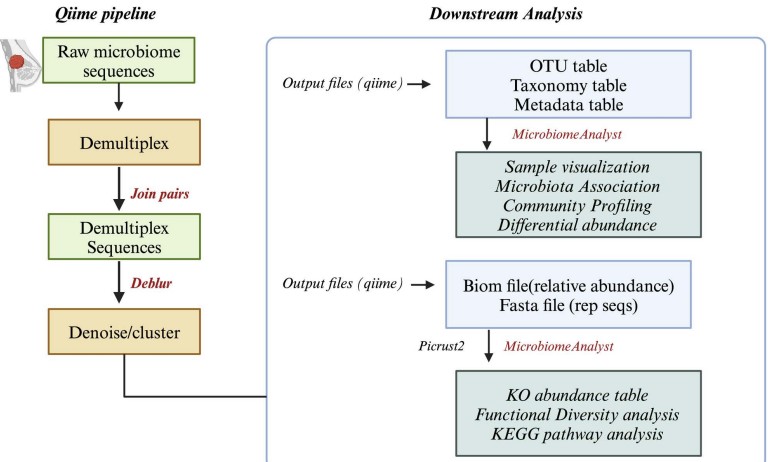

**Fig 1. Schematic Workflow of the Study** A) The entire process, from raw fastq sequences to various outputs (Feature table, Taxonomy table, and representative sequences), is detailed using the Qiime2 pipeline. B) Results from the Qiime2 pipeline were imported into MicrobiomeAnalyst for further analysis. C) Functional diversity and KEGG pathway analyses were conducted using MicrobiomeAnalyst and the R-package MicrobiomeProfiler.

annotation, community profiling (beta and alpha diversity), visual exploration of bacterial species bar plots based on phylum and genus level, and differential abundance testing.

The taxonomic exploration at the phylum and genus levels was shown based on relative abundance. Using statistical t-tests and experimental elements such as circumstances (case vs. control), the Alpha diversity, which measures variety within the community, was determined based on chao1 (evenness), observed (richness), and Shannon (account for both evenness and richness) [22]. The differences between microbial communities (between samples) are accessed using beta diversity. The dissimilarity matrix can be visualized using Principal Coordinate Analysis (PCoA) and calculated using the Bray-Curtis method's distance metric (compositional-based). Permutation ANOVA, or PERMANOVA, is the statistical technique utilized in the beta-diversity analysis [23]. The core microbiome analysis was carried out to determine the core taxa based on sample prevalence and relative abundance. It can be carried out at several taxonomy levels. The Pearson Correlation Coefficient method can be used to determine the correlation between various taxa.

## Differential abundance and pathway analysis

The statistical (EdgeR) [24] and machine learning (LEfSe- Linear Discriminant Analysis: Effect size and Random forest) [25,26] algorithms were utilized to analyze the differential abundance. An ensemble of decision trees is used in the machine learning method known as "random forest classification" to generate predictions. A majority vote or average of the individual guesses determines the predictions made individually by each tree in the forest. Combining the predictions of many models decreases overfitting and increases accuracy, making this a valuable strategy for classification tasks [27].

The functional annotation was done using the Galaxy tool based on the PiCRUST2 pipeline [28]. Research sequences, such as OTUs and ASVs, can be arranged into a reference tree using EPA-NG and GAPPA for sequence placement. Next, the Castor R program predicts the genome for every study sequence in the hidden state prediction. The functional diversity of KEGG metabolism and COG categories using Microbiomeanalyst under the option of SDP. It is possible to stratify the generated metagenome profiles according to the contributing sequences. The microbiome-disease association was performed using the reference database disbiome. Finally, pathway abundances are predicted using metagenome profiles (KEGG pathway). The disease association and KEGG pathway were performed using MicrobiomeProfiler (R-package). The associated microbiome-host genes were referred to through the HOMINID database [29].

## Results

### Summary description of samples

To investigate fecal microbiome differences between DT and CFC samples, 33 metagenome-sequenced samples were sourced from public databases. Post-denoising and filtering, an equal number of samples were processed. Specifically, 19 samples (57.6%) were categorized as CFC, while 14 samples (42.4%) were classified as DT. The distribution of these samples is illustrated in Fig 2A. The total number of Operational Taxonomic Units (OTUs) identified was 301. A detailed description of the samples is provided in Table 1. Fig 2B offers an overview of library sizes, facilitating the identification of potential outliers by detailing the total read count for each sample. Features with low counts were excluded to enhance the robustness of subsequent statistical analyses. Phylogenetic analysis was conducted using distance metrics between the two conditions, with results visualized in a dendrogram (Fig 2C). Rarefaction curve analysis, which plots the number of detected species against the total number of samples, was employed to assess species diversity within the samples. This method aids in estimating species richness, with the curve typically plateauing as sampling effort increases, indicating that most species have been detected. The results of the rarefaction curve analysis are depicted in Fig 2D.

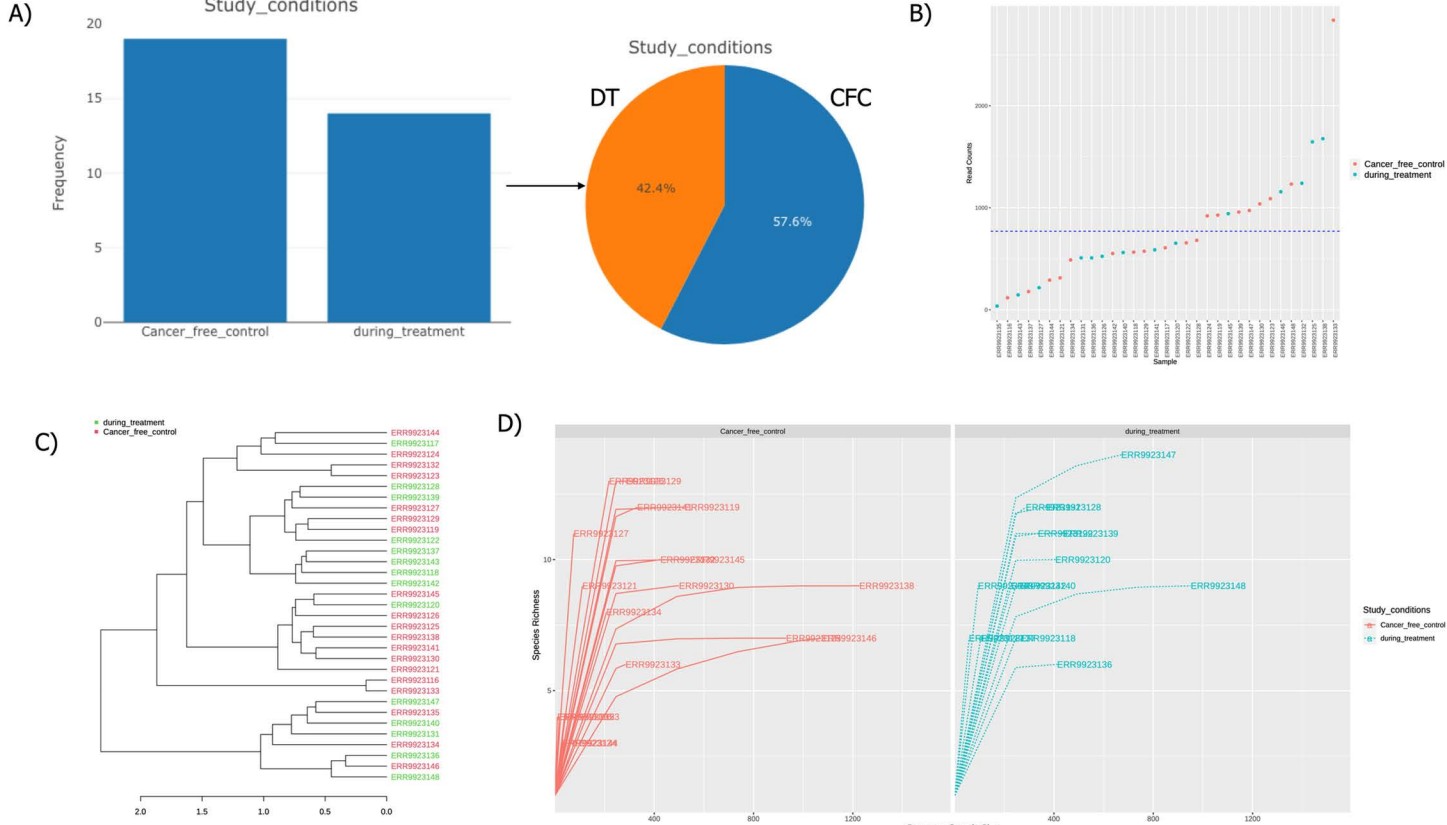

**Fig 2. Summary Description of Samples** A) The distribution of DT and CFC samples is illustrated using bar plots and pie charts. B) An overview of library size, showing the total number of reads per sample, with the x-axis representing the total number of samples and the y-axis indicating read counts. C) Samples are categorized and displayed based on cancer-free control (CFC) and during-treatment (DT) groups. D) Rarefaction curves compare species richness and sequence sampling size between CFC and DT samples, with the y-axis indicating species richness and the x-axis showing sequence sample size. Orange represents the cancer-free control group, and blue represents the during-treatment group.

## Microbiota association of CFC and DT samples

The relative abundance of microbiota in the CFC and DT sample groups was analyzed. The annotation enabled further subdivision and comparison of OTU data based on their abundance across various taxonomic levels. Targeted 16S rRNA gene sequencing was employed to generate stacked horizontal bar plots, illustrating the community's taxonomic composition at both the phylum and genus levels for DT and CFC samples. At the phylum level, *Firmicutes* were more abundant in both sample groups. *Proteobacteria* were the second most abundant phylum, particularly in DT samples. At the genus level, *unclassified_Enterobacteriaceae* were enriched in both the overall and CFC samples, whereas *Bacteroides* were more abundant in DT samples. Fig 3 presents the microbiota composition in actual proportions for the entire sample set and for BC and control samples at both the phylum and genus levels.

## Alpha and beta diversity of CFC and DT samples

Alpha diversity, which measures the variety within a community, was assessed using richness (count) and evenness (distribution) metrics. Specifically, Chao1 (evenness), observed (richness), and Shannon indices (considering both evenness and richness) were computed. Box plots illustrated the disparities between the abundance of the rarest and most numerous species. Chao1 indices revealed that microbial diversity in the CFC group was lower than in the DT group, with a P value of 0.24371. Conversely, the observed and Shannon indices indicated higher microbial diversity in the CFC group compared to the DT group, with P values of 0.48171 and 0.37888, respectively. However, none of these indices showed statistically significant differences in alpha diversity. Detailed results are presented in Fig 4A. Beta diversity, comparing the diversity between two groups, was evaluated using the Bray–Curtis distance metric and permutational MANOVA (PERMANOVA). The microbial composition of each group was visualized using a Principal Coordinate Analysis (PCoA) map (Fig 4B). Due to the limited sample size, no statistically significant beta diversity was observed. Samples closer together in PCoA plots shared similar microbial communities. The two coordinates (PCo1 and PCo2) in the PCoA diagram accounted for 9.2% and 8% of the variation, respectively. The F-value, R-squared, and P-value for the PERMANOVA analysis were

**Table 1. Summary description of the samples.**

| Features | Category |
| --- | --- |
| Accession_ID | PRJEB54599 |
| Sample-type | Fecal |
| Sequencing platform | Illumine MiSeq |
| No of samples | 33 [CFC −19; DT −14] |
| Target region | V4 |
| Data type | OTU abundance table |
| Denoise_method | Deblur |
| OTU annotation | SILVA |
| OTU number | 301 |
| OTU with greater than two counts | 216 |
| Number of experimental factors | 1 |
| Total read counts | 25409 |
| Average counts per sample | 769 |
| Maximum counts per sample | 2838 |
| Minimum counts per sample | 37 |
| Number of samples in metadata | 33 |
| Number of samples in the OTU table | 33 |
| Number of samples processed | 33 |

1.0031, 0.03, and 0.469. Fig 4C displays the 3D PCA plot of these two groups, with the three principal component variations being PC1 (9.2%), PC2 (8%), and PC3 (7%).

## Core microbiome and correlation analysis

Based on sample prevalence and relative abundance, core microbiome analysis identifies core taxa or features that maintain a consistent composition across various sample groups. This analysis is performed at different taxonomic levels. Figs 5A and 5B illustrate the microbiome features at the phylum and genus levels. A correlation study was conducted at various taxonomic levels to identify relationships between taxa. The hierarchical clustering of microbiota in the two distinct groups was examined using relative abundance at the phylum and genus levels, applying the Euclidean clustering distance technique. The results are presented in Figs 5C and 5D. Specifically, a phylum-level correlation study was visualized in a network view and is shown in Fig 6A. Key microbiota at the phylum level are depicted in a boxplot (Fig 6B) using relative abundance. Similarly, a genus-level correlation study was conducted (Fig 6C), with the respective microbiota at the genus level also represented in a boxplot (Fig 6D). These boxplots highlight the major microbiota at both the phylum and genus levels. The analysis algorithm used in this study is the Pearson correlation coefficient, with a significance threshold of P ≤ 0.05.

## Differential abundance and pathway analysis

Differential abundance analysis between the DT and CFC groups was conducted using EdgeR, revealing significant increases and decreases in specific microbiota. Detailed results are provided in Table 2. Biomarker analysis employed

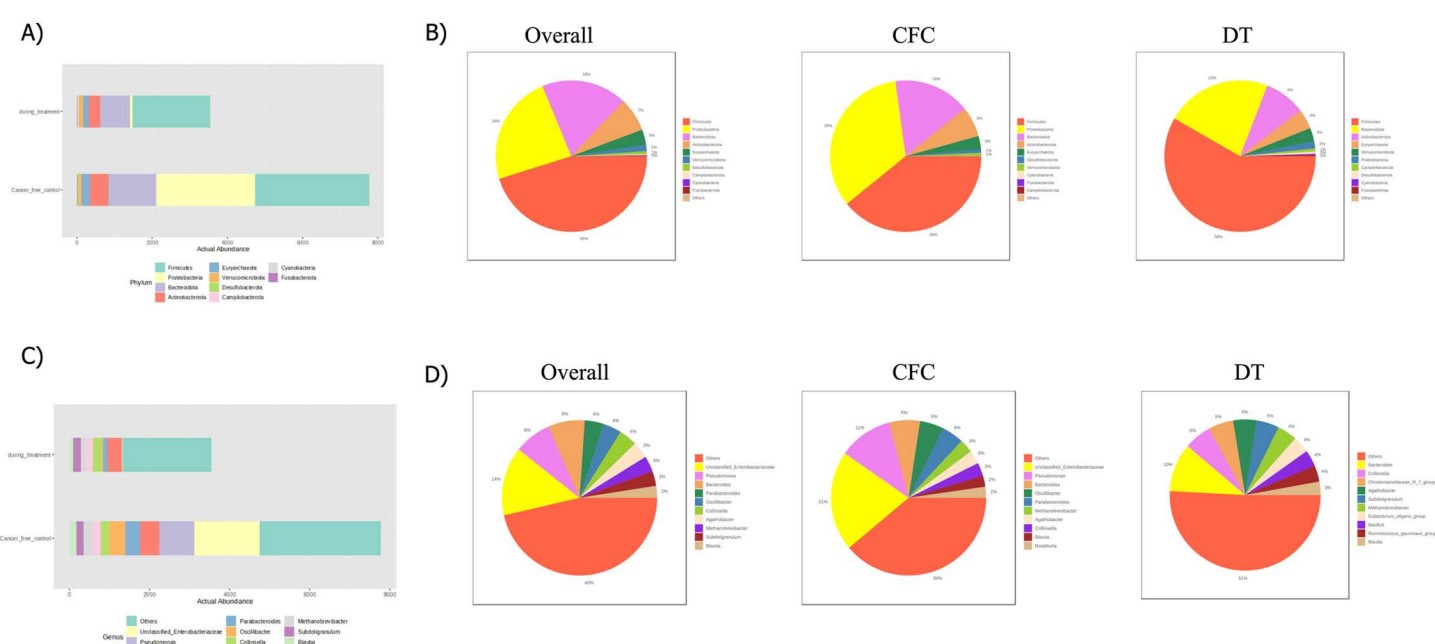

**Fig 3. Microbiota Association of DT and CFC Samples** A) The relative abundance of microbiota associations at the phylum level for DT and CFC samples is shown, with the y-axis indicating actual abundance and the x-axis showing the total number of samples. B) Microbiota associations at the phylum level are categorized into Overall, CFC, and DT samples based on microbiota percentage. C) The relative abundance of microbiota associations at the genus level for CFC and DT samples, with the y-axis indicating actual abundance and the x-axis showing the total number of samples. D) Microbiota associations at the genus level are categorized into Overall, CFC, and DT samples based on microbiota percentage. These results were obtained using MicrobiomeAnalyst.

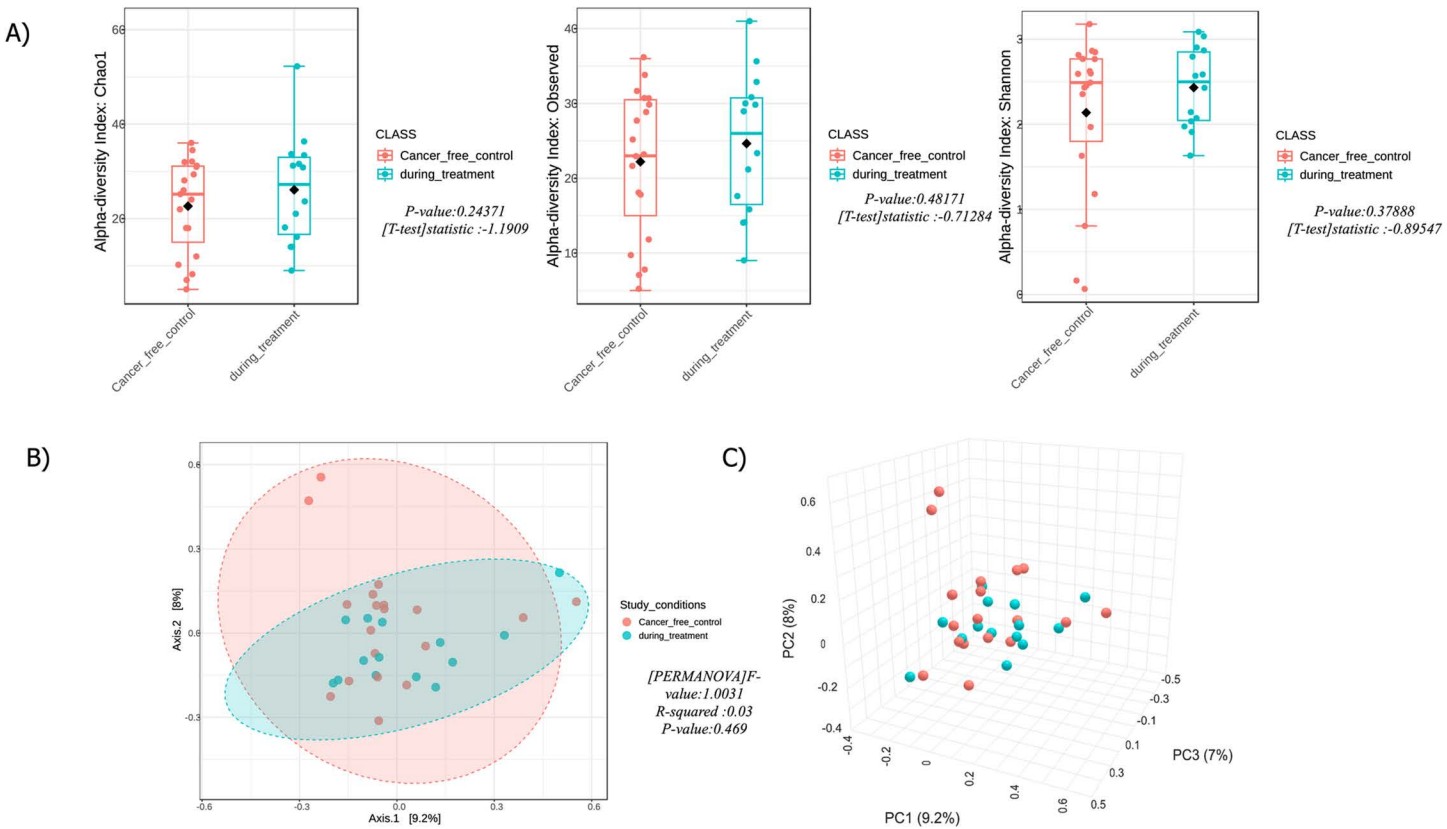

**Fig 4. Alpha and Beta Diversity of DT and CFC Samples** A) Boxplot representing Alpha Diversity's operational taxonomic units (OTUs) of observed species, with the y-axis displaying several groups and the x-axis displaying observed species (OTUs). Measures include chao1, observed, and Shannon. B) Beta-diversity estimate at the phylum level, based on Bray-Curtis metrics (PCoA), showing variations in microbial communities between DT and CFC samples. C) 3D PCA plot of DT and CFC samples. These results were obtained using MicrobiomeAnalyst.

two distinct techniques: random forest and LEfSe. The LEfSe assessment evaluated the microbial abundance of the fecal microbiome in both groups, highlighting differences at various taxonomic levels with a threshold LDA score of ±5.0, as shown in Fig 7A. Key features were predicted and prioritized based on their contribution to classification accuracy, with findings presented in Fig 7B. Functional annotations were performed using the PiCRUST2 pipeline and analyzed with the Galaxy tool. The functional diversity profile was assessed using the KO abundance table as input. Functional diversity and COG categories between CFC and DT groups are depicted in Figs 7C and 7D. Both groups exhibited similar metabolic profiles with higher amino acid, carbohydrate, and nucleotide metabolism abundances. COG categories also showed similar functions with higher abundances in amino acid transport and metabolism and nucleotide transport and metabolism. Differential KO analysis using EdgeR identified 606 KOs as statistically significant (P ≤ 0.05). Detailed information on differential KOs is provided in S2 Table. Microbiome-disease associations were analyzed with a significance threshold of P ≤ 0.05 using the disbiome reference database. Chemotherapy or recovery in BC patients can lead to side effects affecting the brain and other organs, with associated diseases including autism spectrum disorder, multiple sclerosis, and chronic kidney disease. Differentially expressed KOs were used as input for MicrobiomeProfiler to identify critical metabolic pathways enriched between CFC and DT groups, referencing the KEGG database. Key pathways included two-component systems, tyrosine metabolism and propionate metabolism. Associated microbiome-host genes

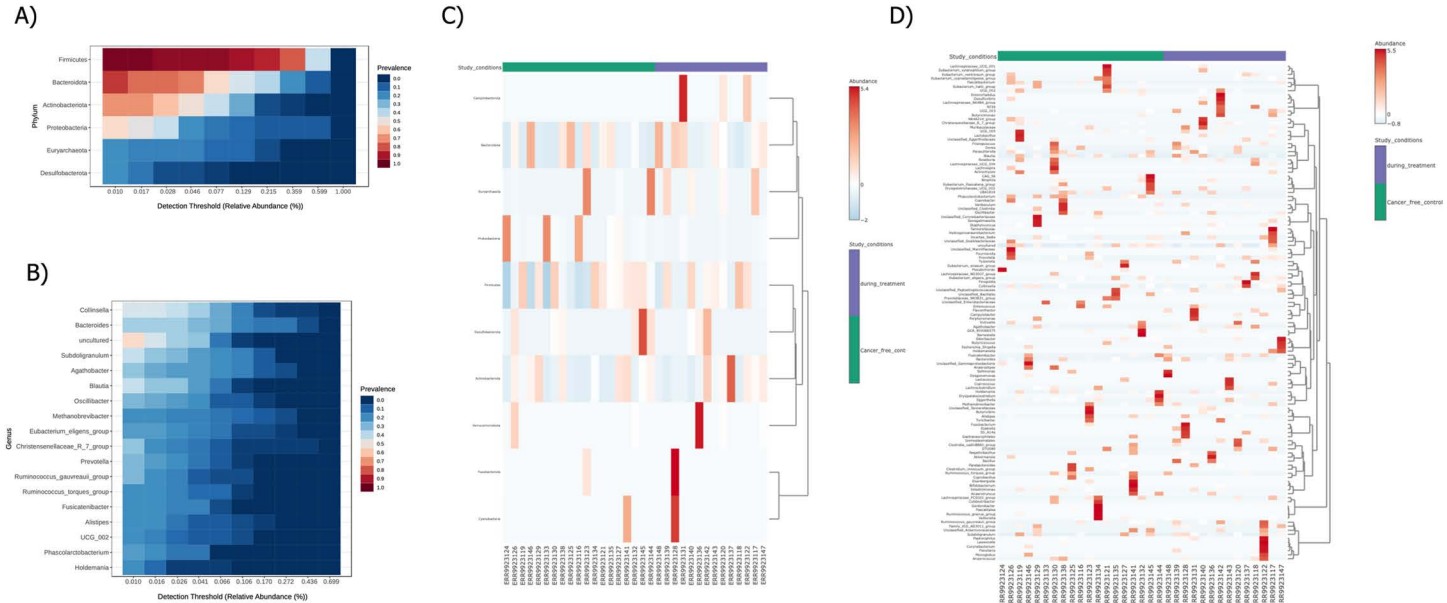

**Fig 5. Core Microbiome and Clustering Analysis of DT and CFC Samples Core microbiome analysis of DT and CFC samples, showing relative abundance** A) Phylum level B) Genus level Relative patterns of high-abundance features of DT and CFC samples at: C) Phylum level D) Genus level.

were referenced through the HOMINID database, with detailed results in Table 3. Notably, the genus *Ruminococcus* was associated with various genes such as *PDZRN3*, *COCH*, and *SOX8*; among these *SOX8* plays significant roles in chemoresistance.

## Discussion

The gut microbiome, comprising trillions of microorganisms residing in the gastrointestinal tract, plays a crucial role in modulating the body's response to chemotherapy [30,31]. Chemotherapy impacts the gut microbiome in several ways, including alterations in microbiome composition, gastrointestinal side effects, drug metabolism, and immune modulation. Chemotherapy can significantly alter the gut microbiome's composition and diversity, often reducing beneficial bacteria and increasing potentially harmful bacteria [32,33]. These changes can contribute to common gastrointestinal side effects of chemotherapy, such as nausea, diarrhea, and mucositis [34]. The gut-brain axis, a bidirectional communication system linking the gut and the brain, can be influenced by changes in the gut microbiome through neurotransmitter production, immune modulation, and regulation of systemic inflammation [35]. The gut microbiome can metabolize chemotherapy drugs, potentially altering their efficacy [36,37]. Protective measures against chemotherapy toxicity include maintaining gut barrier function and leveraging microbial metabolites. A healthy gut microbiome helps preserve the integrity of the gut barrier, protecting against chemotherapy-induced damage and preventing the translocation of harmful bacteria and toxins into the bloodstream [38]. Specific microbial metabolites, such as short-chain fatty acids (SCFAs), have anti-inflammatory and protective effects that can mitigate some chemotherapy side effects [39]. Using 16S rRNA sequencing data, 33 fecal microbiome samples from public sources were compared between DT (14) and CFC (19) samples, targeting the V4 region. Both sample groups showed higher relative abundance of *Firmicutes* and *Proteobacteria*, which are suggested to play significant roles in cancer due to their high abundance. *Bacteroides* were more abundant in both samples at the genus level, while the *Ruminococcus* genus was exclusively present in DT samples. *Bacteroides*, a genus of gram-negative, anaerobic bacteria commonly found in the human gastrointestinal tract, play a significant role in maintaining gut health and are involved in metabolic processes [40,41]. *Bacteroides* and other gut microbiota can influence the

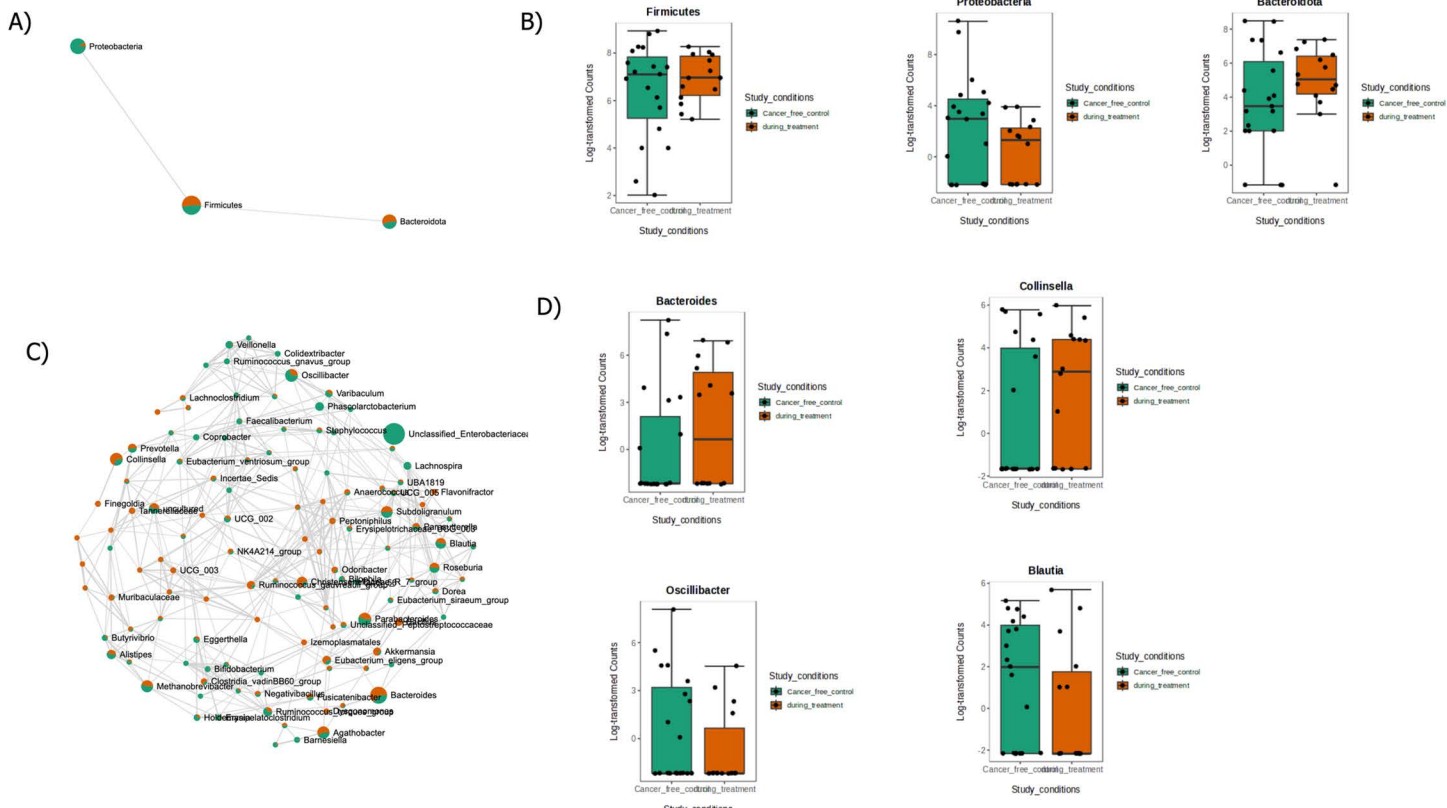

**Fig 6. Correlation Analysis of DT and CFC Samples Network view correlation analysis of DT and CFC sample data:** A) Phylum-level correlation network B) Phylum-level boxplot showing highly abundant microbiome C) Genus-level correlation network D) Boxplot showing high abundance of microbiota by genus.

body's response to cancer treatment by interacting with chemotherapy drugs, potentially affecting their metabolism, efficacy, and toxicity [42]. *Bacteroides* can cause infections as opportunistic pathogens, particularly in immunocompromised patients undergoing chemotherapy [43]. These infections can be significant complications, as chemotherapy suppresses the immune system, increasing vulnerability to bacterial infections. *Ruminococcus*, a genus of gram-positive, anaerobic bacteria in the gut, also affects drug metabolism, immune modulation, and gut health side effects [44,45]. Both genera play significant roles in BC progression and development.

There are no significant changes in alpha and beta diversity between CFC and DT samples, consistent with findings by Bilenduke et al. (2022) [14]. Core microbiota at the phylum and genus levels were evaluated, and clustering analysis of CFC and DT samples was performed. Correlation analysis plotted the microbiota at the phylum and genus levels in a holistic network with statistical significance. Differential abundance of OTUs was assessed using EdgeR, revealing significant increases or decreases in specific microbial species under certain conditions or treatments. Four microbiota showed increased abundance between these sample groups based on logFC: *Bacillus*, *Ruminococcus*, *Allistipes*, and *Eubacterium_ruminantium_group*. Conversely, ten microbiota exhibited decreased abundance: *Unclassified_Enterobacteriaceae*, *Anaerotruncus*, *Pseudomonas*, *Phascolarctobacterium*, *Lachnospira*, *Eubacterium_xylanophilum_group*, *Veillonella*, *Oceanobacillus*, *Dialister*, and *Akkermansia*. Biomarker analysis of DT and CFC groups was conducted using LEfSe and random forest classification, identifying essential microbiota enriched in both approaches. Studies suggest these microbiota are strongly associated with BC. Notably, *Ruminococcus* and *Allistipes* were highly associated with both methods

**Table 2. Differential abundance analysis between DT and CFC sample groups using EdgeR.**

| Microbiome | log2FC | log CPM | Pvalues |
|---|---|---|---|
| *Unclassified_Enterobacteriaceae* | −6.3103 | 16.155 | 1.24E-06 |
| *Anaerotruncus* | −5.5838 | 15.469 | 3.68E-06 |
| *Pseudomonas* | −5.3217 | 15.386 | 7.46E-06 |
| *Bacillus* | 3.1117 | 13.163 | 7.55E-06 |
| *Phascolarctobacterium* | −3.1011 | 13.393 | 2.85E-05 |
| *Ruminococcus* | 2.3461 | 14.2 | 0.0001566 |
| *Alistipes* | 2.1956 | 13.221 | 0.00087358 |
| *Lachnospira* | −2.6502 | 13.156 | 0.00088443 |
| *Eubacterium_ruminantium_group* | 2.2451 | 13.021 | 0.0010143 |
| *Ezakiella* | 1.7825 | 12.485 | 0.0014583 |
| *Jonquetella* | 1.6859 | 12.517 | 0.0014894 |
| *Erysipelotrichaceae_UCG_006* | 1.8984 | 12.609 | 0.0017497 |
| *UCG_003* | 1.551 | 12.624 | 0.0018634 |
| *Butyrivibrio* | 1.9099 | 12.808 | 0.0019498 |
| *Eubacterium_xylanophilum_group* | −2.3144 | 13.006 | 0.0030325 |
| *Veillonella* | −2.2475 | 13.12 | 0.0035098 |
| *Unclassified_Tannerellaceae* | 1.3781 | 12.338 | 0.0035305 |
| *Oceanobacillus* | −2.0526 | 12.744 | 0.0042958 |
| *Flavonifractor* | 1.5082 | 12.485 | 0.0043554 |
| *Gastranaerophilales* | −1.9812 | 12.707 | 0.0052848 |
| *Not_Assigned* | 1.7982 | 13.788 | 0.0056141 |
| *Intestinimonas* | 1.4298 | 12.454 | 0.0064162 |
| *Dialister* | −2.1477 | 13.55 | 0.007047 |
| *Fenollaria* | 1.4108 | 12.349 | 0.0070974 |
| *Coprococcus* | 1.4105 | 12.349 | 0.007204 |
| *Eggerthella* | −1.5004 | 12.797 | 0.011606 |
| *Eubacterium_ventriosum_group* | 1.6407 | 14.021 | 0.013166 |
| *Eubacterium_eligens_group* | 1.6906 | 13.411 | 0.013576 |
| *Odoribacter* | 1.3923 | 12.771 | 0.018166 |
| *Colidextribacter* | −1.4508 | 12.456 | 0.019446 |
| *Dysgonomonas* | 1.3035 | 12.447 | 0.020252 |
| *Unclassified_Peptostreptococcaceae* | 1.2398 | 12.56 | 0.021136 |
| *Akkermansia* | −2.5092 | 16.027 | 0.021769 |
| *Lachnospiraceae_NK4B4_group* | 1.0746 | 12.245 | 0.024322 |
| *Holdemania* | 1.4102 | 12.778 | 0.025924 |
| *Muribaculaceae* | 1.2563 | 12.597 | 0.026536 |
| *Faecalitalea* | −1.3327 | 12.405 | 0.027231 |
| *Lachnospiraceae_FCS020_group* | 1.1707 | 12.686 | 0.030997 |
| *Peptoniphilus* | 1.2908 | 12.736 | 0.031029 |
| *Holdemanella* | 0.98046 | 12.216 | 0.032956 |
| *Christensenellaceae_R_7_group* | 1.5557 | 13.849 | 0.033703 |
| *Erysipelatoclostridium* | −1.3997 | 12.703 | 0.035639 |
| *Tannerellaceae* | 1.1555 | 12.388 | 0.035835 |
| *Alloprevotella* | −1.1588 | 12.425 | 0.042072 |
| *Eubacterium_fissicatena_group* | 1.3239 | 12.816 | 0.043991 |
| *Bifidobacterium* | −1.2373 | 12.459 | 0.05183 |

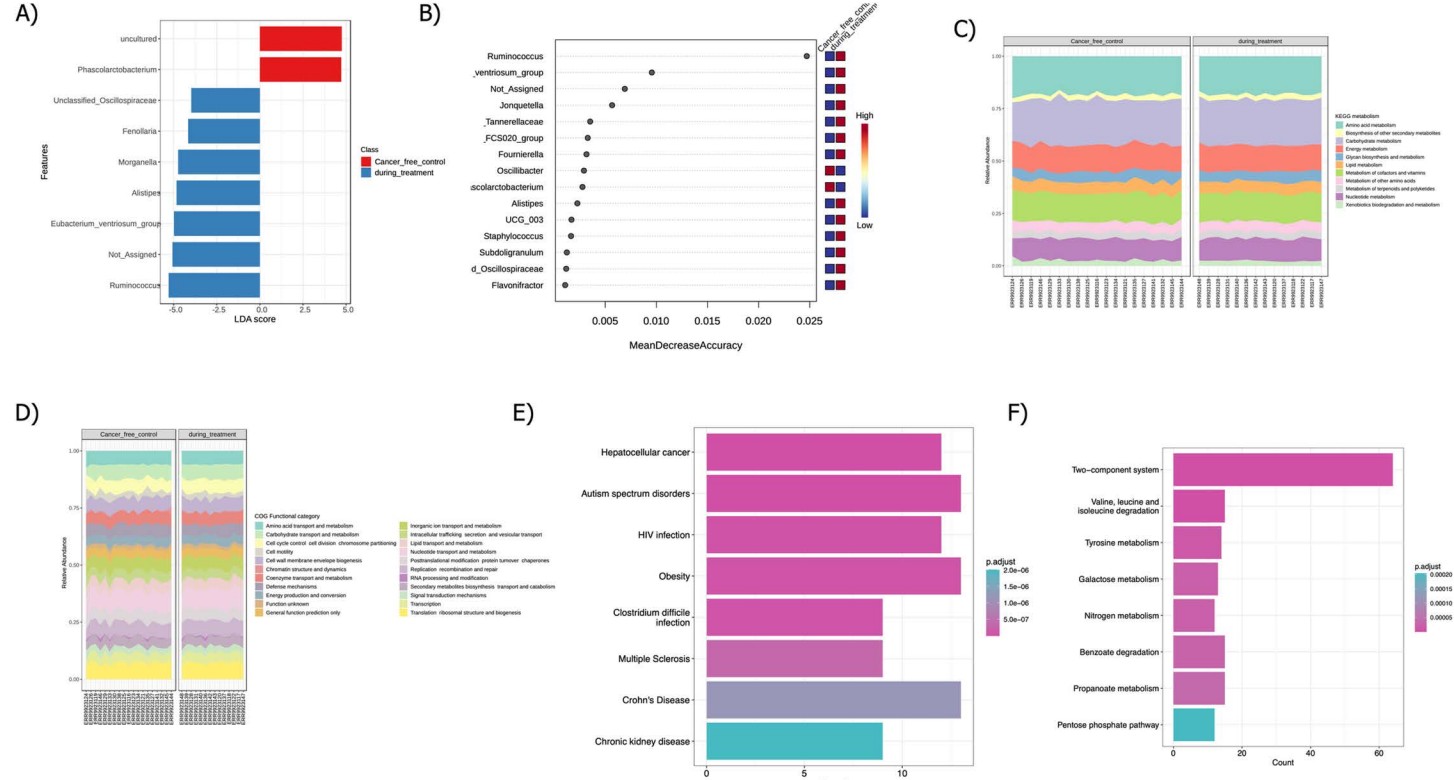

**Fig 7. Biomarker and Pathway Analysis** A) Fecal microbiome abundance in DT and CFC sample groups assessed using LEfSe, with enriched bacteria (genus) linked to CFC (red) and DT (blue) groups depicted in a bar plot. B) Biomarker analysis evaluated using random forest classification of DT and CFC samples, with the x-axis denoting Mean decrease accuracy and the y-axis representing features (microbiota). Functional diversity profiling based on DT and CFC groups: C) KEGG metabolism D) COG functional category E) Disease association analysis of CFC and DT sample groups F) KEGG pathway enrichment using MicrobiomeProfiler with adj P-value ≤ 0.05.

**Table 3. Microbiome-host genetic association.**

| Taxon Set | Enriched in this study | Gene | Total | Hits | Expect | P value |
|---|---|---|---|---|---|---|
| *Clostridiaceae; Dialister; Veillonella; Lentisphaerae* | *Dialister; Veillonella* | *ABCC8* | 4 | 2 | 0.264 | 0.0231 |
| *Collinsella; Coprococcus; Oscillospira; Phascolarctobacterium* | *Coprococcus;Phascolarctobacterium* | *BMPER* | 4 | 2 | 0.264 | 0.0231 |
| *Coriobacteriales; Odoribacter; Ruminococcus; Pasteurellales* | *Odoribacter; Ruminococcus* | *COCH* | 4 | 2 | 0.264 | 0.0231 |
| *Bacteroidales; Bacilli; Lachnospira; Ruminococcus* | *Lachnospira; Ruminococcus* | *PDZRN3* | 4 | 2 | 0.264 | 0.0231 |
| *Finegoldia; Peptoniphilus; Coprococcus; Comamonadaceae; Moraxella* | *Peptoniphilus; Coprococcus;* | *FAM134A* | 5 | 2 | 0.33 | 0.0369 |
| *Prevotellaceae; Bacillus; Clostridiales; Moryella; Veillonella* | *Bacillus;Veillonella* | *MAP1S* | 5 | 2 | 0.33 | 0.0369 |
| *Porphyromonadaceae; Cyanobacteria; Carnobacteriaceae; Butyrivibrio; Ruminococcus; Alcaligenaceae* | *Butyrivibrio; Ruminococcus* | *SOX8* | 6 | 2 | 0.396 | 0.0532 |

[3,46,47]. Functional diversity profiling, KEGG metabolism, and COG functional categories were analyzed to visualize essential functions in CFC and DT samples. Disease-microbiome association analysis highlighted significant disease associations related to brain functions enriched by chemotherapy treatment. Key genera associated with BC treatment were identified by analyzing microbiome-host genetic associations using the HOMINID database. For instance, Ruminococcus is associated with *COCH*, *PDZRN3*, and *SOX8* genes. The gut microbiome can influence host gene expression, including transcription factors like *SOX8*, through mechanisms such as microbial metabolite production acting as signaling molecules. *SOX8* is critical in developing chemoresistance in cancer cells by regulating gene expression, promoting EMT, maintaining cancer stem cell properties, and modulating apoptotic pathways. Understanding *SOX8's* contributions to chemoresistance can aid in developing new therapeutic strategies to enhance chemotherapy effectiveness and improve cancer patient outcomes [48–50].

*Ruminococcus*, a genus of gut microbiota, plays a pivotal role in the degradation of complex carbohydrates and the synthesis of short-chain fatty acids (SCFAs), which are integral to metabolic and gastrointestinal health [51,52]. Although no direct evidence links *Ruminococcus* to the *SOX8* gene, potential indirect interactions remain plausible. The *SOX8* gene, a member of the SOX transcription factor family, is implicated in critical biological functions, including neurogenesis, spermatogenesis, and oncogenesis [53]. A plausible link might lie within the gut-brain axis, where the microbiota influence neural function and gene regulation, possibly modulating SOX8 activity [35,54]. Furthermore, SCFAs such as butyrate—metabolic byproducts of Ruminococcus—are known to influence gene expression through epigenetic mechanisms, suggesting a potential indirect effect on SOX8. Additionally, the immunomodulatory properties of Ruminococcus may reshape gene regulatory pathways in tissues where *SOX8* is expressed [55,56]. While direct evidence of an association between *Ruminococcus* and *SOX8* is currently absent, these hypothetical pathways underline the need for further research into their potential interactions. Fig 8 mentions the mechanism of the interconnected gut-brain axis. Overall, a better understanding of the gut-brain axis in chemotherapy could lead to improved strategies for managing the cognitive, emotional, and physical side effects of cancer treatment.

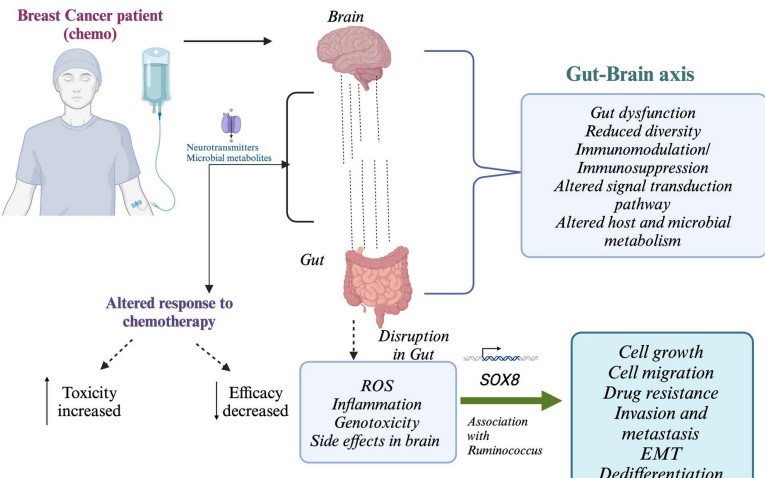

**Fig 8. Mechanism of Gut-Brain Axis-The gut-brain axis underscores the importance of holistic cancer treatment due to the interdependence of these systems.**

## Conclusion

The study identified *Firmicutes* and *Proteobacteria* as the predominant phyla in the breast microbiome of DT and CFC samples, with *Bacteroides* and *Ruminococcus* being the most common genera. Both alpha and beta diversity analyses revealed an imbalance in microbial diversity distribution. Correlation studies highlighted significant genera linked to specific critical microbiota. Biomarkers and differential abundance of OTUs were determined using statistical and machine-learning techniques. The richness and composition of the microbiome can be influenced by host genetic factors, underscoring the importance of the microbiome-host genetic link in BC and chemoresistance. In conclusion, the gut microbiome represents a promising frontier in cancer therapy, potentially enhancing treatment efficacy, reducing toxicity, and personalizing cancer care.

## Study limitations

This investigation is constrained by its limited sample size of 33 participants (19 CFC and 14 DT) and the absence of longitudinal data. These factors diminish the reliability and generalizability of the findings. Specifically, a small sample size undermines statistical power, heightening the risk of Type I (false-positive) and Type II (false-negative) errors, which complicates the identification of genuine biological variations. Moreover, the reduced sample size restricts the applicability of the results to broader populations and increases vulnerability to biases, such as selection bias and sampling variability. The absence of longitudinal data further impedes the ability to monitor temporal dynamics, making it challenging to evaluate disease progression, treatment responses, or the stability of biomarkers over time. These limitations underscore the critical need for future research incorporating larger sample sizes and longitudinal methodologies to bolster statistical rigor, refine interpretability, and enhance the validity of conclusions.

## Future directions

Future research endeavors aim to enhance our comprehension of the gut microbiome's influence on chemotherapy and to explore therapeutic strategies for optimizing patient outcomes. This includes investigating specific microbial taxa and metabolites involved in chemotherapy response, developing microbiome-modulating therapies, and incorporating microbiome analysis into clinical practice.

## Supporting information

**S1 Table. Metadata information of the samples used in this study.**
(XLSX)

**S2 Table. Comprehensive details of differentially represented KEGG Orthologs (KOs), including KO identifiers, functional descriptions, fold changes, and statistical significance values observed in the analysis.**
(XLSX)

## Acknowledgments

The authors would like to take this opportunity to thank the management of Vellore Institute of Technology (VIT), Vellore, India, for providing the necessary facilities and encouragement to carry out this work.

## Author contributions

**Conceptualization:** Tamizhini Loganathan, George Priya Doss C.

**Investigation:** Tamizhini Loganathan, George Priya Doss C.

**Methodology:** Tamizhini Loganathan, George Priya Doss C.

**Project administration:** George Priya Doss C.

**Supervision:** George Priya Doss C.

**Validation:** George Priya Doss C.

**Visualization:** Tamizhini Loganathan.

**Writing – original draft:** Tamizhini Loganathan.

**Writing – review & editing:** George Priya Doss C.

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
