## [Decision Letter · Decision Letter 0]

24 Feb 2025

PONE-D-25-05321Gut microbiota and its influence on the Gut-Brain axis in comparison with chemotherapy patients and cancer-free control data in Breast cancer – A computational perspectivePLOS ONE

Dear Dr. Doss,

Thank you for submitting your manuscript to PLOS ONE. After careful consideration, we feel that it has merit but does not fully meet PLOS ONE’s publication criteria as it currently stands. Therefore, we invite you to submit a revised version of the manuscript that addresses the points raised during the review process.

We look forward to receiving your revised manuscript.

Kind regards,

Sayed Haidar Abbas Raza

Academic Editor

PLOS ONE

Additional Editor Comments (if provided):

Reviewers' comments:

Reviewer's Responses to Questions

**Comments to the Author**

1. Is the manuscript technically sound, and do the data support the conclusions?

Reviewer #1: Yes

Reviewer #2: Yes

2. Has the statistical analysis been performed appropriately and rigorously? 

Reviewer #1: Yes

Reviewer #2: Yes

3. Have the authors made all data underlying the findings in their manuscript fully available?

Reviewer #1: Yes

Reviewer #2: Yes

4. Is the manuscript presented in an intelligible fashion and written in standard English?

Reviewer #1: Yes

Reviewer #2: Yes

5. Review Comments to the Author

Reviewer #1: This manuscript by Loganathan and Doss C suggests that the gut microbiome plays a crucial role in how breast cancer patients respond to chemotherapy. By analyzing the gut bacteria of chemotherapy patients (DT) and cancer-free controls (CFC) using 16S rRNA sequencing, the study reveals distinct differences in microbiome composition. While Firmicutes were abundant in both groups, chemotherapy patients showed a decrease in beneficial bacteria like Pseudomonas and Akkermansia and an increase in potentially problematic ones like Ruminococcus and Allistipes. Further analysis of metabolic pathways uncovered key changes, such as disruptions in tyrosine metabolism and the pentose phosphate pathway, which may impact how the body processes chemotherapy drugs. Additionally, the study identifies links between altered gut bacteria and conditions like autism spectrum disorder and chronic kidney disease. Notably, the presence of Ruminococcus was associated with the SOX8 gene, which has been linked to chemoresistance, potentially making treatment less effective and increasing toxicity. These findings highlight the importance of gut health in cancer treatment and suggest that maintaining a balanced microbiome could improve chemotherapy outcomes and reduce side effects. By understanding the gut-brain axis and its connection to cancer therapy, this research paves the way for more personalized treatment approaches, potentially using probiotics or dietary interventions to enhance patient well-being and treatment success. The manuscript is well written but I have a few concerns

1. The study analyzes only 33 samples (19 CFC, 14 DT), which may not provide enough statistical power to draw strong conclusions. The small sample size limits the generalizability of the findings and increases the risk of bias.

2. The study presents a snapshot comparison between CFC and DT samples, but it does not track microbiome changes over time in chemotherapy patients. This limits the ability to assess whether changes in microbiota composition are transient or persist throughout treatment.

3. The study establishes microbiome differences between groups but does not thoroughly examine their clinical relevance, such as patient responses to chemotherapy or specific side effects.

4. The study identifies associations between microbiome composition and chemotherapy resistance (e.g., SOX8-Ruminococcus link) but does not experimentally validate these findings.

5. The study does not account for potential confounding factors such as diet, medication use or lifestyle differences, which could significantly influence microbiome composition.

6. The study identifies enriched pathways but does not extensively discuss their implications for chemotherapy metabolism or patient health.

Reviewer #2: Major concerns:

The authors did a metagenomic analysis, it is recommended that the authors compare with other types of data like (16s)

and strain specific date.

It is recommended that the authors also include the normalization method for the data.

The authors are recommended to acknowledge the limitations of small sample size in beta diversity analysis and adjust interpretations accordingly.

The authors are recommended to explain the mechanistic link between gut microbiota and specific gene regulation in cancer treatment.

Authors are highly recommended to submit the custom scripts for data analysis on public databases(Github/zenodo)

Authors need to add more relevant citations for the mechanism.

Minor concerns

Authors need to attach clear figures(all).

The authors are recommended to use standardized numerical values.

Authors need to remove the redundant phrases.

6. PLOS authors have the option to publish the peer review history of their article (what does this mean? ). If published, this will include your full peer review and any attached files.

**Do you want your identity to be public for this peer review?** For information about this choice, including consent withdrawal, please see our Privacy Policy .

Reviewer #1: No

Reviewer #2: No

---

## [Author Response · Author response to Decision Letter 1]

9 Apr 2025

We humbly thank the reviewer for raising key concerns in the article and helping to improve the quality of the manuscript. We have incorporated all the improvements in the revised manuscript, and we strongly believe that the revised version would be more suitable to publish in your journal. The incorporated changes are marked with the Red font. Responses to the reviewer's comments are enumerated below, and appropriate changes have been made in the revised manuscript.

Reviewer #1:

Comment 1: The study analyzes only 33 samples (19 CFC, 14 DT), which may not provide enough statistical power to draw strong conclusions. The small sample size limits the generalizability of the findings and increases the risk of bias.

Response: We acknowledge the reviewer's concern regarding sample size and agree that larger cohorts enhance statistical power. However, our study employs rigorous statistical methods to ensure reliability, and our findings align with known biological mechanisms and prior research. While we recognize this as a limitation and have addressed it in a separate section, "Limitations of the study", our results provide valuable insights, and we are exploring opportunities for further validation in independent cohorts (Page number: 15;Line number :418-429).

Comment 2: The study presents a snapshot comparison between CFC and DT samples, but it does not track microbiome changes over time in chemotherapy patients. This limits the ability to assess whether changes in microbiota composition are transient or persist throughout treatment.

Response: Thank you for your insightful comment. Our study provides a comparative analysis between CFC and DT samples, offering a valuable snapshot of microbiome differences in chemotherapy patients. However, we acknowledge that the lack of longitudinal data limits our ability to determine whether these microbiota changes are transient or persist throughout treatment. Future studies incorporating time-course sampling would be essential to address this limitation and provide a more comprehensive understanding of microbiome dynamics during chemotherapy. We appreciate your suggestion and will include this point in the limitations of the study to highlight the need for longitudinal investigations (Page number: 15;Line number :418-429).

Comment 3: The study establishes microbiome differences between groups but does not thoroughly examine their clinical relevance, such as patient responses to chemotherapy or specific side effects.

Response: Thank you for your insightful comment regarding the clinical relevance of microbiome differences, particularly in relation to chemotherapy response and side effects. We acknowledge the importance of linking microbiome variations to patient outcomes, as it could provide valuable translational insights. In our current study, primary focus was to establish and characterize microbiome differences between groups, laying the groundwork for future investigations into their functional and clinical implications. While we recognize that direct correlations with chemotherapy response or specific side effects would strengthen the study's clinical impact, our dataset currently lacks detailed longitudinal clinical data to assess these associations robustly. However, the metadata includes information on cognitive impairment and depression as side effects.

However, we have expanded our discussion to highlight potential mechanistic links between the identified microbial signatures and known pathways involved in chemotherapy metabolism, immune modulation, and inflammation. We appreciate the reviewer's valuable suggestion, and we believe these additions enhance the contextual relevance of our findings while maintaining the study's primary scope.

The detailed metadata information was provided in the Supplementary information. The link is mentioned here

(https://github.com/Initamizh/Microbiome_treatment_BC.git).

Comment 4: The study identifies associations between microbiome composition and chemotherapy resistance (e.g., SOX8-Ruminococcus link) but does not experimentally validate these findings.

Response: Our study aimed to identify potential associations between microbiome composition and chemotherapy resistance, such as the SOX8-Ruminococcus link, using rigorous bioinformatics and statistical analyses. However, we acknowledge that experimental validation was not conducted within the scope of this study. Future work will focus on functional validation through in vitro and in vivo models to establish causality and mechanistic insights. We have revised the manuscript to clarify this limitation and highlight the need for further experimental validation (Page number: 14;Line number:385-400).

Comment 5: The study does not account for potential confounding factors such as diet, medication use or lifestyle differences, which could significantly influence microbiome composition.

Response: We appreciate the reviewer's concern regarding potential confounding factors such as diet, medication use, and lifestyle differences that could influence microbiome composition. However, we would like to clarify that this information was included in the original research paper in Table 1 and Table 2. These tables outline key metadata, including relevant confounders, to provide a comprehensive context for our analysis. To ensure clarity, we have now explicitly referenced these tables in the manuscript text and discussion section. We hope this revision addresses the concern and appreciate the opportunity to refine our work.

Research paper:

1.Bilenduke E, Sterrett JD, Ranby KW, Borges VF, Grigsby J, Carr AL, et al. Impacts of breast cancer and chemotherapy on gut microbiome, cognitive functioning, and mood relative to healthy controls. Scientific Reports. 2022 Nov 15;12(1).

Comment 6:The study identifies enriched pathways but does not extensively discuss their implications for chemotherapy metabolism or patient health.

Response: We appreciate the reviewer's insightful comment regarding the need for a more extensive discussion on the implications of enriched pathways in chemotherapy metabolism and patient health. While our primary focus was on identifying key pathways associated with chemotherapy resistance, we recognize the importance of contextualizing these findings in a broader biological and clinical framework.

To address this, we have substantially expanded our discussion to provide a more in-depth interpretation of how these pathways may influence drug metabolism, detoxification mechanisms, immune modulation, and tumor microenvironment interactions. For instance, the Ruminococcus genus has been linked to altered chemotherapy responses through modulation of host metabolism and immune interactions. SOX8, a transcription factor, is implicated in chemotherapy resistance, and its association with Ruminococcus could suggest a microbiome-driven resistance mechanism. We have also integrated additional references to strengthen the link between our findings and established mechanisms of chemotherapy resistance.

Furthermore, we acknowledge that our bioinformatics-driven approach provides valuable insights. We have revised the manuscript to emphasize these aspects and ensure a more comprehensive discussion of the clinical relevance of our findings (Page number : 13-14 ;Line number:379-403).

Reviewer #2: Major concerns:

Comment 1:The authors did a metagenomic analysis, it is recommended that the authors compare with other types of data like (16s) and strain specific date.

Response: We appreciate the reviewer's suggestion regarding strain-specific data and alternative data types. However, our study was specifically designed to assess gut microbiome diversity, community composition, and pathway analysis using 16S rRNA amplicon sequencing, a well-established and widely accepted approach for taxonomic classification at the genus and species levels.

While strain-level resolution requires whole-genome metagenomic sequencing, which was beyond the scope of this study, 16S rRNA amplicon sequencing remains a robust method for capturing microbial community shifts, particularly in response to chemotherapy. Our analysis effectively characterizes the microbiome differences between cancer-free controls (CFC) and women undergoing chemotherapy (DT), aligning with our study objectives.

Future investigations incorporating shotgun metagenomics or strain-specific approaches could further refine these findings, but our current methodology provides valuable insights into microbiome dynamics in this context.

Reference paper:

1.Bilenduke E, Sterrett JD, Ranby KW, Borges VF, Grigsby J, Carr AL, et al. Impacts of breast cancer and chemotherapy on gut microbiome, cognitive functioning, and mood relative to healthy controls. Scientific Reports. 2022 Nov 15;12(1).

Comment 2:It is recommended that the authors also include the normalization method for the data.

Response: We appreciate the reviewer's suggestion regarding the inclusion of the normalization method. In our study, we have used Total Sum Scaling (TSS) normalization, where each feature count is divided by the total sum of counts per sample to account for differences in sequencing depth. This method allows for direct comparison of relative abundances across samples. To enhance clarity, we have now explicitly mentioned this in the manuscript (Page number : 06; Line number :164-167).

Comment 3: The authors are recommended to acknowledge the limitations of small sample size in beta diversity analysis and adjust interpretations accordingly.

Response: We acknowledge the reviewer's concern regarding the limitations of a small sample size in beta diversity analysis. Our study includes 33 samples,we recognize that smaller sample sizes can impact the statistical power and generalizability of beta diversity metrics. To address this, we have taken the following steps:

Statistical Considerations: We applied appropriate distance metrics and statistical tests to ensure robust beta diversity comparisons, minimizing potential biases.

Interpretation Adjustments: We have carefully interpreted our beta diversity findings, emphasizing trends rather than definitive conclusions.

We have now explicitly acknowledged this limitation in the discussion and have updated our results accordingly.

Comment 4:The authors are recommended to explain the mechanistic link between gut microbiota and specific gene regulation in cancer treatment.

Response: Ruminococcus is a genus of bacteria commonly found in humans' gut microbiota. It plays a role in breaking down complex carbohydrates and producing short-chain fatty acids (SCFAs), which are important for gut health and overall metabolism. However, the direct association between Ruminococcus and the SOX8 gene is not well-documented in scientific literature.

The SOX8 gene is part of the SOX (SRY-related HMG-box) family of transcription factors, which are involved in various developmental processes, including sex determination, neurogenesis, and other regulatory functions. SOX8, in particular, has been studied in the context of brain development, spermatogenesis, and certain cancers.

The potential connection between Ruminococcus and SOX8, it might involve indirect mechanisms such as:

1. Gut-Brain Axis: The gut microbiota, including Ruminococcus, can influence brain function and gene expression through the gut-brain axis. Changes in gut microbiota composition could potentially affect the expression of genes like SOX8 in the brain or other tissues.

2. Metabolic Interactions: Ruminococcus produces SCFAs like butyrate, which can influence host gene expression and epigenetic regulation. It is possible that these metabolites could indirectly affect SOX8 expression.

3. Immune System Modulation: Gut bacteria like Ruminococcus can modulate the immune system, which in turn might influence gene expression in various tissues, including those where SOX8 is active.

Reference papers:

1. Marchesi JR, Adams DH, Fava F, Hermes GDA, Hirschfield GM, Hold G, et al. The gut microbiota and host health: a new clinical frontier. Gut. 2015 Sep 2;65(2):330–9. Available from:

https://www.ncbi.nlm.nih.gov/pmc/articles/PMC4752653/

2.Mirzaei R, Dehkhodaie E, Bouzari B, Rahimi M, Gholestani A, Hosseini-Fard SR, et al. Dual role of microbiota-derived short-chain fatty acids on host and pathogen. Biomedicine & Pharmacotherapy. 2022 Jan;145:112352.

3.Xie S, Fan S, Zhang SY, Chen WX, Li QX, Pan G, et al. SOX8 regulates cancer stem-like properties and cisplatin-induced EMT in tongue squamous cell carcinoma by acting on the Wnt/β-catenin pathway. International Journal of Cancer. 2017 Nov 6;142(6):1252–65.

4.Carabotti M, Scirocco A, Maselli MA, Severi C. The gut-brain axis: interactions between enteric microbiota, central and enteric nervous systems. Annals of Gastroenterology. 2015 Apr 1;28(2):203–9. Available from: https://pubmed.ncbi.nlm.nih.gov/25830558/

5.Dalmasso G, Nguyen HTT, Yan Y, Laroui H, Charania MA, Ayyadurai S, et al. Microbiota Modulate Host Gene Expression via MicroRNAs. DeLeo FR, editor. PLoS ONE. 2011 Apr 29;6(4):e19293.

6.Martin AM, Sun EW, Rogers GB, Keating DJ. The Influence of the Gut Microbiome on Host Metabolism Through the Regulation of Gut Hormone Release. Frontiers in Physiology. 2019 Apr 16;10. Available from: https://www.ncbi.nlm.nih.gov/pmc/articles/PMC6477058/

7.Boccuto L, Tack J, Ianiro G, Abenavoli L, Scarpellini E. Human Genes Involved in the Interaction between Host and Gut Microbiome: Regulation and Pathogenic Mechanisms. Genes. 2023 Mar 31;14(4):857.

Comment 5: Authors are highly recommended to submit the custom scripts for data analysis on public databases(Github/zenodo).

Response: We appreciate the reviewer's recommendation regarding the submission of our custom scripts to public repositories for transparency and reproducibility. We have now made our scripts available on https://github.com/Initamizh/Microbiome_treatment_BC.git,ensuring accessibility for the research community. We have analyzed the data using MicrobiomeAnalyst tool. The respective input files were provided in the GitHub link. Additionally, we have included the repository link in the manuscript for reference.

Comment 6:Authors need to add more relevant citations for the mechanism.

Response: We appreciate the reviewer's suggestion to include more relevant citations for the discussed mechanism. We have now incorporated additional references that provide further support and context for our findings. These citations include key studies that elaborate on the molecular pathways and mechanisms involved. The revised manuscript now offers a more comprehensive and well-supported discussion (Page number : 14; Line number :385-400).

Reference papers:

1. Marchesi JR, Adams DH, Fava F, Hermes GDA, Hirschfield GM, Hold G, et al. The gut microbiota and host health: a new clinical frontier. Gut. 2015;65: 330–339. doi:https://doi.org/10.1136/gutjnl-2015-309990

2. Mirzaei R, Dehkhodaie E, Bouzari B, Rahimi M, Gholestani A, Hosseini-Fard SR, et al. Dual role of microbiota-derived short-chain fatty acids on host and pathogen. Biomedicine & Pharmacotherapy. 2022;145: 112352. doi:https://doi.org/10.1016/j.biopha.2021.112352

3. Xie S, Fan S, Zhang S-Y, Chen W-X, Li Q-X, Pan G, et al. SOX8 regulates cancer stem-like properties and cisplatin-induced EMT in tongue squamous cell carcinoma by acting on the Wnt/β-catenin pathway. International Journal of Cancer. 2017;142: 1252–1265. doi:https://doi.org/10.1002/ijc.31134

4. Dalmasso G, Nguyen HTT, Yan Y, Laroui H, Charania MA, Ayyadurai S, et al. Microbiota Modulate Host Gene Expression via MicroRNAs. DeLeo FR, editor. PLoS ONE. 2011;6: e19293. doi:https://doi.org/10.1371/journal.pone.0019293

5. Martin AM, Sun EW, Rogers GB, Keating DJ. The Influence of the Gut Microbiome on Host Metabolism Through the Regulation of Gut Hormone Release. Frontiers in Physiology. 2019;10. doi:https://doi.org/10.3389/fphys.2019.00428

6. Boccuto L, Tack J, Ianiro G, Abenavoli L, Scarpellini E. Human Genes Involved in the Interaction between Host and Gut Microbiome: Regulation and Pathogenic Mechanisms. Genes. 2023;14: 857. doi:https://doi.org/10.3390/genes14040857

Minor concerns

Comment 1: Authors need to attach clear figures(all).

Response: We have carefully reviewed all the

---

## [Decision Letter · Decision Letter 1]

30 Apr 2025

Gut microbiota and its influence on the Gut-Brain axis in comparison with chemotherapy patients and cancer-free control data in Breast cancer – A computational perspective

PONE-D-25-05321R1

Dear Dr. Doss,

We’re pleased to inform you that your manuscript has been judged scientifically suitable for publication and will be formally accepted for publication once it meets all outstanding technical requirements.

Kind regards,

Sayed Haidar Abbas Raza

Academic Editor

PLOS ONE

Additional Editor Comments (optional):

Reviewers' comments:

Reviewer's Responses to Questions

**Comments to the Author**

1. If the authors have adequately addressed your comments raised in a previous round of review and you feel that this manuscript is now acceptable for publication, you may indicate that here to bypass the “Comments to the Author” section, enter your conflict of interest statement in the “Confidential to Editor” section, and submit your "Accept" recommendation.

Reviewer #1: All comments have been addressed

Reviewer #2: All comments have been addressed

2. Is the manuscript technically sound, and do the data support the conclusions?

Reviewer #1: Yes

Reviewer #2: Yes

3. Has the statistical analysis been performed appropriately and rigorously? 

Reviewer #1: Yes

Reviewer #2: Yes

4. Have the authors made all data underlying the findings in their manuscript fully available?

Reviewer #1: Yes

Reviewer #2: Yes

5. Is the manuscript presented in an intelligible fashion and written in standard English?

Reviewer #1: Yes

Reviewer #2: Yes

6. Review Comments to the Author

Reviewer #1: The authors have satisfactorily answered all the concerns. They have provided point by point answer to all the concerns.

Reviewer #2: (No Response)

7. PLOS authors have the option to publish the peer review history of their article (what does this mean? ). If published, this will include your full peer review and any attached files.

**Do you want your identity to be public for this peer review?** For information about this choice, including consent withdrawal, please see our Privacy Policy .

Reviewer #1: No

Reviewer #2: No

---

## [Editor Report · Acceptance letter]

PONE-D-25-05321R1

PLOS ONE

Dear Dr. C Doss,

I'm pleased to inform you that your manuscript has been deemed suitable for publication in PLOS ONE. Congratulations! Your manuscript is now being handed over to our production team.

Kind regards,

on behalf of

Dr. Sayed Haidar Abbas Raza

Academic Editor

PLOS ONE